# Quartet II: Accurate LLM Pre-Training in NVFP4 by Improved Unbiased Gradient Estimation

**Andrei Panferov** [1]  **Erik Schultheis** [1]  **Soroush Tabesh** [1]  **Dan Alistarh** [1,2]

## Abstract

The NVFP4 lower-precision format, supported in hardware by NVIDIA Blackwell GPUs, promises to allow, for the first time, end-to-end fully-quantized pre-training of massive models such as LLMs. Yet, existing quantized training methods still sacrifice some of the representation capacity of this format in favor of more accurate unbiased quantized gradient estimation by stochastic rounding (SR), losing noticeable accuracy relative to standard FP16 and FP8 training. In this paper, improve the state of the art for quantized training in NVFP4 via a novel unbiased quantization routine for micro-scaled formats, called MS-EDEN, that has more than 2x lower quantization error than SR. We integrate it into a novel fully-NVFP4 quantization scheme for linear layers, called Quartet II. We show analytically that Quartet II achieves consistently better gradient estimation across all major matrix multiplications, both on the forward and on the backward passes. In addition, our proposal synergizes well with recent training improvements aimed specifically at NVFP4. We further validate Quartet II on end-to-end LLM training with up to 1.9B parameters on 38B tokens. We provide kernels for execution on NVIDIA Blackwell GPUs with up to 4.2x speedup over BF16. Our code is available at https://github.com/IST-DASLab/Quartet-II.

## 1. Introduction

The computational cost of training state-of-the-art foundation models has been increasing at a roughly exponential pace, putting into question the sustainability of the area, e.g. (Amodei & Hernandez, 2018; Sevilla et al., 2022). Pre-training modern Transformer-based foundation values is dominated by dense matrix multiplications (GEMMs), e.g.

the linear projections in attention and MLPs, and so, reducing the precision of these GEMMs is one of the most direct levers for lowering end-to-end training costs.

This motivation has driven a steady progression of mixed-precision training recipes, from FP16/BF16 to FP8 (Micikevicius et al., 2022), and now toward 4-bit *microscaling* floating point formats such as MXFP and NVFP. In these formats, values are stored in a 4-bit floating-point encoding, but each small block is accompanied by a higher-precision, e.g. FP8, scale, preserving dynamic range while enabling tensor-core acceleration. Recent GPU accelerators provide native support for such formats, with 2-4x throughput gains over FP8 for individual matmuls (NVIDIA, 2024).

The key challenge is to retain FP16/FP8-quality optimization while performing most operations at 4-bit precision (Xi et al., 2023; Chmiel et al., 2024). At this scale, naive quantization leads to divergence over long pre-training runs. Emerging work on stable FP4 native training (Tseng et al., 2025; Castro et al., 2025; Chmiel et al., 2025) has converged on two guiding principles. First, the *forward pass* should seek to maximize representation capacity by minimizing the quantization error of activations and weights, typically measured via mean-square error (MSE). Second, the *backward pass* is especially sensitive to bias: as such, biased gradient estimators can accumulate systematic error over many steps, making *unbiased* (or carefully controlled) gradient quantization essential for stable convergence. These insights underpinned NVIDIA's first end-to-end NVFP4 pre-training recipe (NVIDIA et al., 2025) and subsequent refinements, including forward-pass scale selection heuristics (Cook et al., 2025) and improved NVFP4 stability mechanisms (Chen et al., 2025b). Yet, current state-of-the-art FP4 recipes still drop significant accuracy relative to FP8 and FP16.

**Contributions.** In this paper, we improve the current state of the art for NVFP4 native training by revisiting the question of unbiased gradient estimation for the NVFP4 microscaling format. Surprisingly, we show that the prevailing prior solution, element-wise FP4 stochastic rounding (SR), can be significantly improved. We do so by introducing a new unbiased quantization routine for microscaling formats, called MicroScaling EDEN (*MS-EDEN*), that reduces quantization error by moving the stochasticity from individual FP4 values to the microscale factors, while retaining prov-

[1]Institute of Science and Technology Austria [2]Red Hat AI. Correspondence to: Dan Alistarh <Dan.Alistarh@ist.ac.at>.

*Proceedings of the $43^{rd}$ International Conference on Machine Learning*, Seoul, South Korea. PMLR 306, 2026. Copyright 2026 by the author(s).

able unbiasedness in expectation. Based on MS-EDEN, we build *Quartet II*, a fully-NVFP4 linear-layer computation graph that combines (i) a high-capacity forward pass using native NVFP4 scaling augmented with the "Four-over-Six" scale selection heuristic (Cook et al., 2025), with (ii) an unbiased backward pass based on MS-EDEN and efficient inner-dimension randomized block rotations. We provide an analytic comparison showing that Quartet II yields consistently improved gradient estimation across the major matrix multiplications in transformer training, and we validate these improvements in end-to-end LLM pre-training. Finally, we provide kernels enabling efficient execution on NVIDIA Blackwell GPUs, making the proposed recipe practical at scale. In summary, our contributions are as follows:

- A new unbiased quantization primitive called **MS-EDEN** tailored to microscaling FP4 formats that substantially reduces quantization error relative to FP4 stochastic rounding while remaining hardware-compatible and efficient.

- A fully-NVFP4 linear-layer training graph called **Quartet II** that combines improved forward-pass quantization with improved unbiased backward-pass quantization (MS-EDEN), yielding better gradient estimates.

- **Empirical validation**: we perform extensive ablations and end-to-end accuracy validation via training runs showing consistent accuracy improvements over prior NVFP4 recipes.

- **Efficient kernels**: we show that our scheme is efficiently implementable on the NVIDIA Blackwell generation of GPUs, with up to 4.2x speedup vs BF16.

## 2. Related Work

**Lower-precision training.** Low-precision training is a long-standing direction in deep learning, e.g. (Courbariaux et al., 2015; Esser et al., 2019; Panferov et al., 2025a; Micikevicius et al., 2022; Hernández-Cano et al., 2025). Early demonstrations of 4-bit training and 4-bit matrix multiplications focused on INT4, and established that careful handling of scaling and outliers can maintain accuracy in constrained regimes (Xi et al., 2023; Chmiel et al., 2024).

**Training in microscaling FP4.** The recent introduction of NVFP4 and MXFP4 microscaling floating point formats (NVIDIA, 2024) has led to renewed interest in this direction. Tseng et al. (2025) investigated having only the backward pass in MXFP4, highlighting how microscaling choices and kernel behavior interact with optimization stability. Castro et al. (2025) and Chmiel et al. (2025) concurrently proposed the first stable fully-quantized training recipes. The former focused on MXFP4 and used a combination of Hadamard rotations and MSE-optimal clipping, providing GPU kernel results, whereas the latter focused on

NVFP4, employing careful RTN quantization and selective stochastic rounding, with larger-scale (1T token) emulated training results. NVIDIA et al. (2025) introduced the first large-scale recipe for NVFP4, leveraging square block quantization, Hadamard rotations on the backward pass, and setting some layers in higher precision. TetraJetV2 (Chen et al., 2025b) enhanced the NVIDIA approach via improved outlier control and oscillation-reduction techniques. The FourOverSix technique (Cook et al., 2025) provided an orthogonal improvement via an MSE-reducing specialized grid selection algorithm.

TetraJet-v2 was proposed by Chen et al. (2025b) as an upgrade over NVIDIA et al. (2025). It introduces a number of corrections to the scheme as well as a number of heuristics to further stabilize it: i) they correct the activations re-quantization in the backward pass to better align with the chain rule and add weigh re-quantization similar to Castro et al. (2025); ii) they introduce intermediate-level FP32 scales and selective outlier channels. The practicality of these format changes is hard to validate, as it requires substantially more complicated kernel support that is not provided by the authors. In light of that, when referencing TetraJet-v2 later in the paper, we will refer to the following GPU-feasible scheme: NVFP4 quantization with RTN without square-block-scales on the forward pass, and SR quantization with RHT on the inner dimension for both GEMMs on the backward pass. We will not, however, re-implement their intermediate FP32 scales or outlier channels. This separates the logical scheme from design decisions that would be difficult to implement in practice.

All the above techniques employ some variant of stochastic rounding (SR) to preserve unbiasedness on the backward pass. We re-consider this choice, and propose a new unbiased gradient estimator (MS-EDEN) which provides significantly better MSE, and validation loss.

**Unbiased quantization and rotations.** Unbiased stochastic quantization is a key technique in distributed optimization (Alistarh et al., 2017; Suresh et al., 2017; Davies et al., 2020), as it leads to convergence guarantees for communication-reduced SGD. Stochastic rounding is the standard unbiased primitive in low-precision training, but can substantially inflate variance at lower bitwidths. EDEN (Vargaftik et al., 2022) combined randomized rotations with a corrective rescaling to obtain (nearly) unbiased estimators in distributed optimization. Yet, this technique is not directly applicable in our setting, as we discuss in Section 3.2. Our MS-EDEN routine addresses this issue by enabling unbiasedness while reducing error relative to SR. More broadly, rotations have also been used for distribution smoothing in the case of weight and activation quantization (Tseng et al., 2024; Ashkboos et al., 2024).

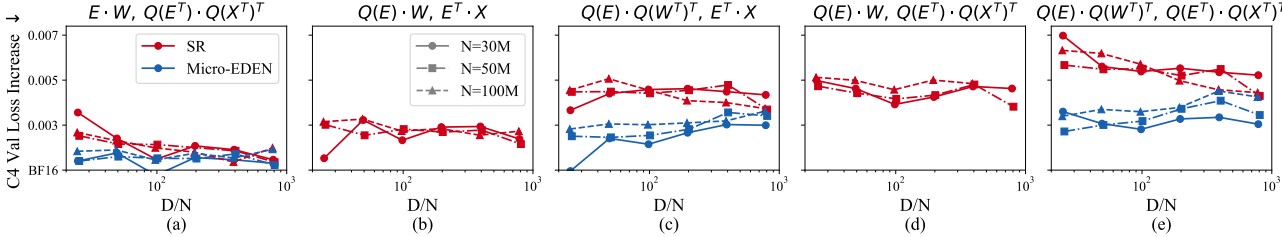

*Figure 1.* Impact of selective NVFP4 backward pass quantization on C4 Validation Loss relative to BF16 pre-training for $N$-parameter Llama-2-like LLMs with $D/N$ tokens-per-parameter. Axis captions indicate which tensors of the two backward pass GEMMs are quantized.

## 3. Backward Pass Quantization

A $d$-dimensional quantization operator $Q$ applied to vectors $\boldsymbol{x}^d \in \mathbb{R}^d$ is usually defined as (possibly stochastic) mapping $Q(\boldsymbol{x}^d, \omega) \to \mathbb{R}^d$ where the argument $\omega \in \Omega$ is given by probability samples used to unbias the result. In practice, users can sample the (pseudo-) randomness $\omega$ reproducibly from its distribution $\Omega$. Then, *unbiasedness* w.r.t. $\omega$ is defined as follows:

$$\forall \boldsymbol{x}^d \in \mathbb{R}^d : \mathbb{E}_\omega \left[ Q(\boldsymbol{x}^d, \omega) \right] = \boldsymbol{x}^d.$$

We will focus on unbiasedness in backward pass quantization for LLM pre-training, where it was shown to be crucial for stable long-term convergence (Chmiel et al., 2024; Tseng et al., 2025; Castro et al., 2025; NVIDIA et al., 2025). Intuitively, this is because consistent bias in gradient estimation can lead to persistently incorrect descent directions.

### 3.1. NVFP4 and Stochastic Rounding

The end goal of quantized training is to achieve higher throughput via the use of specialized low-precision GEMMs; in particular, recent GPUs from NVIDIA and AMD support micro-scaling formats called MXFP4 and NVFP4. The NVFP4 format was shown to yield superior accuracy to MXFP4 (NVIDIA et al., 2025; Egiazarian et al., 2025; Chen et al., 2025a). It represents tensors mapping values to the E2M1 floating point format with two levels of scales: one E4M3 scale per 16 values, and a single FP32 scale per tensor, for range extension. Formally, the quantized representation $Q_{\text{SR}}$ of $\boldsymbol{x}$ becomes

$$Q_{SR}(\boldsymbol{x} \in \mathbb{R}^d, \omega) \to \begin{cases} \boldsymbol{x}^{\text{FP4}} \in \mathbb{R}^d \\ \boldsymbol{x}^{\text{FP8}} \in \mathbb{R}^{d//16} \\ x^{\text{FP32}} \in \mathbb{R} \end{cases} \to \mathbb{R}^d,$$

where $\boldsymbol{x}^{\text{FP4}} \in \mathbb{R}^d$ is the vector of FP4 elements, $\boldsymbol{x}^{\text{FP8}} \in \mathbb{R}^{d//16}$ is the set of group scales, and $x^{\text{FP32}}$ is the scalar global scale. Then, Stochastic Rounding (SR) is defined as:

$$x^{\text{FP32}} = \max_{i=1...d} |\boldsymbol{x}_i| / (6.0 \times \frac{16}{17} \times 448.0),$$

$$\boldsymbol{x}_g^{\text{FP8}} = \text{RTN}_{\text{FP8}} \left( \max_{i=16 \cdot g...16 \cdot g+15} \frac{|\boldsymbol{x}_i|}{x^{\text{FP32}} \times 6.0 \times \frac{16}{17}} \right),$$

$$\boldsymbol{x}_i^{\text{FP4}} = \text{SR}_{FP4} \left( \frac{\boldsymbol{x}_i}{\boldsymbol{x}_{i//16}^{\text{FP8}} \times x^{\text{FP32}}}, \omega \right).$$

Here, $448.0$ is the absolute maximum value representable by FP8, $6.0$ is the absolute maximum value representable by FP4 and $16/17$ is the maximum factor by which $\text{RTN}_{\text{FP8}}$ can increase the underlying values. The latter is necessary to ensure that $-6.0 \le \frac{\boldsymbol{x}_i}{\boldsymbol{x}_{i//16}^{\text{FP8}} \times x^{\text{FP32}}} \le 6.0$, similar to the $3/4$ factor of Tseng et al. (2025) for MXFP4. $\text{SR}_{\text{FP4}}$ is the probabilistic rounding operation w.r.t. randomness $\omega$ which preserves its argument in expectation. Given the choice of constants, stochastic rounding $\text{SR}_{\text{FP4}}$ does not clip its arguments, and the resulting estimation is unbiased, that is:

$$\mathbb{E}_\omega \left[ \boldsymbol{x}_i^{\text{FP4}} \times \boldsymbol{x}_{i//16}^{\text{FP8}} \times x^{\text{FP32}} \right] = \boldsymbol{x}_i.$$

To our knowledge, all existing FP4 training methods (Chmiel et al., 2025; Castro et al., 2025; NVIDIA et al., 2025; Chen et al., 2025b; Tseng et al., 2025) utilize element-wise stochastic rounding for unbiasedness.

### 3.2. EDEN Rescaling: A Theoretically-Justified Solution

A popular tool in the context of LLM quantization is given by randomized rotations such as the Randomized Hadamard Transform (RHT) (Xi et al., 2023; Tseng et al., 2024; Ashkboos et al., 2024; Tseng et al., 2025). An alternative use of the RHT comes from distributed optimization (Suresh et al., 2017; Davies et al., 2020; Vargaftik et al., 2021). One such method is EDEN (Vargaftik et al., 2022), which uses RHT (seeded by the random variable $\omega$) to ensure co-linearity between the high-precision rotated vector $\text{RHT}(\boldsymbol{x}, \omega)$ and the expectation of the quantized vector $Q(\text{RHT}(\boldsymbol{x}, \omega))$. The

**Algorithm 1** MS-EDEN

**Input:** vector $\boldsymbol{x}$, rotation seed $\omega_{\text{RHT}}$, rounding seed $\omega_{\text{SR}}$, grid max $s$

**for** $h$ in range(0, $d$, 128): **do**
$\quad \boldsymbol{x}^{\text{RHT}}_{[h:h+128]} = \text{RHT}(\boldsymbol{x}_{[h:h+128]}, \omega_{\text{RHT}})$
**end for**
$\{\boldsymbol{x}^{\text{FP4}}, \boldsymbol{x}^{\text{FP8}}, x^{\text{FP32}}\} = Q_{\text{RTN}}(\boldsymbol{x}^{\text{RHT}}, s)$
$\boldsymbol{x}^{\text{RTN}} = \boldsymbol{x}^{\text{FP4}} \times \boldsymbol{x}^{\text{FP8}} \times x^{\text{FP32}}$
**for** $h$ in range(0, $d$, 16): **do**
$\quad S_{h//16} = \frac{\langle \boldsymbol{x}^{\text{RHT}}_{[h:h+16]}, \boldsymbol{x}^{\text{RHT}}_{[h:h+16]} \rangle}{\langle \boldsymbol{x}^{\text{RHT}}_{[h:h+16]}, \boldsymbol{x}^{\text{RTN}}_{[h:h+16]} \rangle}$
**end for**
**for** $g$ in range(0, $d//16$): **do**
$\quad \boldsymbol{x}^{\text{FP8}}_g = \text{SR}_{\text{FP8}}\left(S_g \cdot \boldsymbol{x}^{\text{FP8}}_g, \omega_{\text{SR}}\right)$
**end for**
**return** $\{\boldsymbol{x}^{\text{FP4}}, \boldsymbol{x}^{\text{FP8}}, x^{\text{FP32}}\}$

key idea is introducing a *bias correction factor $S$* via:

$$S = \frac{\langle \boldsymbol{x}, \boldsymbol{x} \rangle}{\langle \text{RHT}(\boldsymbol{x}, \omega), Q(\text{RHT}(\boldsymbol{x}, \omega)) \rangle},$$
$$Q_{\text{EDEN}}(\boldsymbol{x}, \omega) = S \cdot Q(\text{RHT}(\boldsymbol{x}, \omega)). \quad (1)$$

Given this construction, the EDEN authors show that, if $d$ is the underlying dimension, then:

$$\lim_{d \to \infty} \mathbb{E}_\omega \left[ \text{RHT}^{-1}\left(Q_{\text{EDEN}}(\boldsymbol{x}, \omega), \omega\right) \right] = \boldsymbol{x},$$

i.e., $Q_{\text{EDEN}}$ yields unbiased estimates in rotated space. In practice, this sequence converges fast enough to be unbiased with RHT performed in groups as small as $d = 64$.

**The Challenge.** Unfortunately, this elegant construction cannot be directly applied to gradient estimation for quantized training. As observed by Castro et al. (2025), the scale correction factor $S$ proposed by EDEN has values in the interval $[0.94, 1.06]$ in practice, requiring a high precision representation for storage. As such, it is incompatible with the coarse compressed scale representation of NVFP4: the minimum relative update that can be accommodated by FP8 scales is $\times 1.0625$. Nor can this be merged into the finer per-tensor scale, as the scaling groups have to be a subset of the rotation groups.

### 3.3. Our Solution: Microscaling EDEN

**Overview.** We now show how to extend the EDEN bias correction given in Equation 1 to the NVFP4 microscaling quantization format. The pseudocode of our procedure is given in Algorithm 1. We first provide an overview, and then discuss some key implementation details.

The procedure processes the input vector $\boldsymbol{x}$ in chunks of e.g. 128 consecutive entries (any multiple of the quantization group size 16 is valid), given rotation and rounding seeds,

*Table 1.* Quadratic error over $\mathcal{N}(0, 1)$ for a number of NVFP4 rounding schemes with native (1x16) or square-block (16x16) scales. Addition of Four Over Six (Cook et al., 2025) is indicated by "+4/6". Highlighted are the chosen schemes for Quartet II forward pass and backward pass.

| Method | Group Size | MSE $\times 10^{-3}$ | Unbiased |
|--------|-----------|---------------------|----------|
| RTN | 1x16 | 9.0 | ✗ |
| **+4/6** | 1x16 | **7.6** | ✗ |
| RTN | 16x16 | 12.4 | ✗ |
| +4/6 | 16x16 | 12.4 | ✗ |
| SR | 1x16 | 23.5 | ✓ |
| +4/6 | 1x16 | 17.5 | ✗ |
| **MS-EDEN** | 1x16 | **9.4** | ✓ |

and a grid scale parameter $s$. First, we perform an RHT of the current chunk, seeded by the corresponding pseudo-randomness $\omega_{\text{RHT}}$. This rotated chunk is then quantized to NVFP4 via *round-to-nearest (RTN)* quantization, yielding substantially lower mean-square-error than standard SR. The second step requires us to address the EDEN scale precision issue. For this, we propose a novel variant that merges the EDEN correction factors $S$ into the group micro-scales via stochastic rounding. The unbiasedness of stochastic rounding guarantees that, in expectation, $S$ is represented exactly, preserving the unbiasedness end-to-end.

**"Unbiased" NVFP4 RTN Quantization.** First, notice that, since EDEN guarantees unbiasedness via randomized rotations and re-scaling, we do not need stochastic rounding (SR) of individual values to FP4. Second, since we not employ SR, we can allow the $Q_{\text{RTN}}$ operation to possibly clip some values. Third, the correction factors might sometimes need to scale $\boldsymbol{x}^{\text{FP8}}$ "up," meaning that we need to raise the range ceiling to accommodate these updates. To accommodate these constraints, we define the clipping RTN NVFP4 quantization scheme $Q_{\text{RTN}}(\boldsymbol{x} \in \mathbb{R}^d, s \in \mathbb{R})$ as follows:

$$x^{\text{FP32}} = \max_{i=1...d} |\boldsymbol{x}_i| / (s \times 256.0),$$
$$\boldsymbol{x}^{\text{FP8}}_g = \text{RTN}_{\text{FP8}}\left(\max_{i=16 \cdot g...16 \cdot g+15} \frac{|\boldsymbol{x}_i|}{x^{\text{FP32}} \times s}\right),$$
$$\boldsymbol{x}^{\text{FP4}}_i = \text{RTN}_{\text{FP4}}\left(\frac{\boldsymbol{x}_i}{\boldsymbol{x}^{FP8}_{i//16} \times x^{\text{FP32}}}\right).$$

Setting the clipping factor $s$ to $6 \times \frac{16}{17}$ or lower makes the scheme non-clipping. We numerically find that $s = \frac{1}{0.93} \times 6 \times \frac{16}{17}$ minimizes the expected MSE over $\mathcal{N}(0, 1)$. We use this factor for the rest of the paper. Additionally, relative to $Q_{\text{SR}}$, FP8 scales are initially capped by 256.0 instead of 448.0 for them not to overflow when applying EDEN correction. The only place where we require stochastic rounding is for the group scales, in order to address the fact that NVFP4 group scales are maintained in E4M3 FP8, which is too coarse to faithfully represent the EDEN rescaling factors.

**Guarantees.** Formally, the quantizer $Q$ needs to satisfy a number of properties for EDEN to be unbiased, such as i) sign-symmetry and ii) $Q(\boldsymbol{x}) \neq 0$. These properties hold for non-under-flowing floating point quantization. Specifically, NVFP4 was shown to have enough range for the scales not to underflow (Egiazarian et al., 2025; Chen et al., 2025a). Based on Theorem 2.1 of Vargaftik et al. (2022) and the properties of stochastic rounding, the following hold:

**Corollary 3.1.** *For all $\boldsymbol{x} \in \mathbb{R}^d$ and scale $s \neq 0$, we have:*

$$\widehat{\boldsymbol{x}} = Q_{\mathrm{MS-EDEN}}(\boldsymbol{x}, \omega_{\mathrm{RHT}}, \omega_{\mathrm{SR}}, s)$$
$$\mathbb{E}_{\omega_{\mathrm{RHT}}, \omega_{\mathrm{SR}}} \mathrm{RHT}^{-1}(\widehat{\boldsymbol{x}}, \omega_{\mathrm{RHT}}) = \boldsymbol{x}.$$

In practice, the inverse rotation $\mathrm{RHT}^{-1}$ does not need to be performed, as it naturally cancels out when MS-EDEN is applied on the inner dimension of a matrix multiplication to both tensors with the same rotation seed $\omega_{\mathrm{RHT}}$. We numerically validate unbiasedness in Appendix A.

**Practical Performance.** In Table 1, we show the MSE over normally-distributed data for various NVFP4 quantizers: round to nearest (RTN) over groups of size 1x16 and 16x16, and Stochastic Rounding (SR) with and without the FourOverSix (4/6) grid heuristic (Cook et al., 2025).

First, observe that SR achieves unbiasedness at the cost of approximately 2.5x increase in MSE over RTN. At the same time, MS-EDEN shows much smaller error increase, improving by more than 2x over SR. We attribute this to the fact that (a) per-element stochastic rounding introduces significant variance that if fully avoided in MS-EDEN, (b) the rescaling with $S$ is small for NVFP4 and (c) stochastic rounding for the 8-bit scales introduces variance an order of magnitude smaller than 4-bit quantization itself.

The reliance on randomized rotations, however, imposes additional limitations: Micro-scaling groups have to be subdivisions of rotation groups. Due to hardware restrictions, this implies that rotations and scale corrections have to be applied on the inner dimension of multiplied tensors. Thus, there are additional considerations about the computation scheme to be made to use MS-EDEN for unbiased gradient estimations in LLMs.

## 4. Forward Pass Quantization

### 4.1. A Representation-Requantization Trade-Off

Beyond the use of stochastic rounding, one consistent feature for the NVIDIA NVFP4 LLM pre-training scheme (NVIDIA et al., 2025) and follow-up work (Cook et al., 2025) is *square-block* quantization of the weight tensor $W$ in the forward pass $Y = XW^T$. This is designed to allow the re-use, without re-quantization, of the quantized tensor in the backward pass operation for computing the

input gradient:

$$\frac{\partial L}{\partial X} \approx Q_{\mathrm{FP4}}(E) \cdot Q_{\mathrm{FP4}}(W^T)^T.$$

This effectively halves the backward pass quantization error for this matrix product, as seen from stochastic rounding (SR) performance gaps in Figure 1 (b,c). This improvement, however, comes at the cost of worse outlier preservation and generally lower representation capacity on the forward pass, due to effectively having a single FP8 scale per 256 FP4 values, instead of per 16 values. The effect on forward pass quantization accuracy can be observed in Figure 2, where square blocks ("16x16gs") consistently lag behind NVFP4 native blocks ("1x16gs") in terms of LLM validation perplexity. This presents a trade-off between gradient estimation quality and model representation capacity chosen by enabling or disabling square-group-quantization, which NVIDIA et al. (2025) resolve towards the former.

We make a different choice here. One first reason is that MS-EDEN requires the application of randomized rotations along the micro-scaling group dimension, i.e., along the inner GEMM dimension. This creates the need to requantize the weight tensor $W$ and activations tensor $X$ in the backward pass. Second, we argue that, even with weight re-quantization, MS-EDEN yields lower error than SR without weight re-quantization, since it has more than 2x lower quadratic error (Table 1). Moreover, this can be seen by comparing SR without weight re-quantization in Figure 1 (d) with MS-EDEN with weight re-quantization in Figure 1 (e), which shows how this finding extrapolates to LLM pre-training (more details in Section 6.1). Thus, MS-EDEN enjoys a better forward pass representation, while improving gradient estimation on the backward pass.

### 4.2. Forward Pass Using "4/6"

Cook et al. (2025) propose Four Over Six ("4/6"), a modification to the NVFP4 quantization algorithm that evaluates two potential scale factors (4.0 and 6.0) for each block of values, and picks the one that yields lower MSE. They combine "4/6" with stochastic rounding on backward pass.

Yet, this combination has a notable correctness issue. In the form proposed, it *does not constitute an unbiased estimation*, as the act of picking a lower MSE scale branch introduces bias, even if both scale branches are individually unbiased via SR. We validate this claim empirically in Appendix A. Consequently, their scheme does not produce unbiased gradient estimations and, as such, we disregard it from the backward pass comparison.

Its usefulness for forward pass, however, is clear. In their original scheme, this idea is not utilized due to the use of square-block-quantization for the weight tensor. We validate this by measuring the quadratic error improvement from "4/6" on $\mathcal{N}(0, 1)$ tensors and report the results in Table 1.

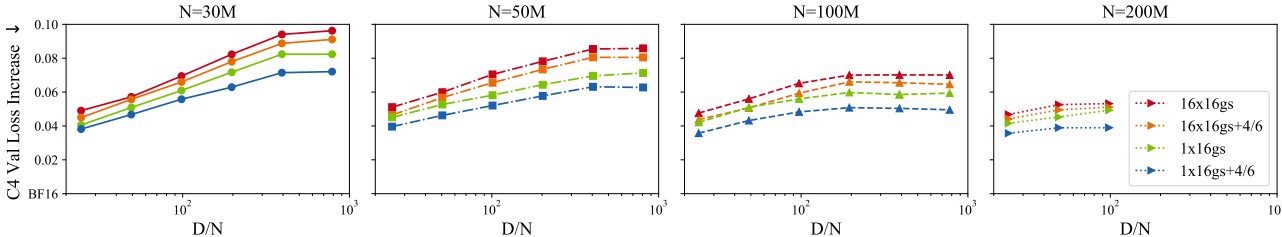

*Figure 2.* NVFP4 Forward Pass C4 Validation Loss Gaps relative to BF16 pre-training for $N$-parameter Llama-2-like LLMs with $D/N$ tokens-per-parameter. "16x16gs" and "1x16gs" indicate whether square block quantization was used or not and "+4/6" indicates whether Four Over Six (Cook et al., 2025) was used.

Moreover, we measure the effect of "4/6" on forward pass QAT and report validation loss increase in Figure 2 (see Section 6.1). One can see how "4/6" positively synergizes with native NVFP4 scales on the forward pass, showing roughly double the improvement compared to square-block-quantization for LLM pre-training.

## 5. The Quartet II Computation Graph

We now put everything together to propose Quartet II, a fully-NVFP4 linear layer computation scheme for LLM pre-training, with unbiased gradient estimation guarantees.

For the **Forward Pass**, Quartet II uses Round-to-Nearest FP4 rounding with native NVFP4 scaling (one FP8 E4M3 scale per 16 elements) and additionally one per-tensor FP32 scale for range extension (NVIDIA et al., 2025). This is augmented by a local scaling level choice for the quantization grid following Cook et al. (2025), which we refer to as "4/6". This deterministic rounding operation is applied to both weights and activations in forward pass, and allows native NVFP4 multiplication using tensor cores on Blackwell NVIDIA GPUs. The quantized weights and activations are saved for their use on the backward pass.

For the **Backward Pass**, a group RHT rotation matrix is first generated using pseudo-randomness. The saved quantized weights and activations are then de-quantized, transposed and then re-quantized with MS-EDEN along with the tensors $E$ and $E^T$ to yield unbiased estimations of the corresponding tensors. These quantized tensors are then multiplied in NVFP4 tensor cores. The product outputs need no further processing, as the rotations cancel out along the inner GEMM dimensions. They are then fed to the optimizer steps and further in back-propagation.

**The Computational Graph** is illustrated in Figure 3. This scheme is designed to improve upon the TetraJet-v2 scheme (Chen et al., 2025b) and, by extension, the NVIDIA recipe (NVIDIA et al., 2025). One key difference is the replacement of SR quantization with MS-EDEN on the backward pass and the addition of finer "4/6" (Chen et al., 2025a) scale selection on the forward pass.

## 6. Experimental Validation and Extensions

### 6.1. Llama-Family Model Pre-Training

We now provide experimental validation for Quartet II by ablating its components on LLM pre-training. Specifically, we train Transformer models (Vaswani et al., 2023) following the Llama 2 (Touvron et al., 2023) architecture on language modeling loss on samples from the C4 dataset (Dodge et al., 2021) using Adam (Kingma & Ba, 2017) with cosine LR schedule (Loshchilov & Hutter, 2017). We train models with 30M, 50M, 100M and 200M parameters with data-to-parameter ratios in 25, 50, 100, 200, 400 and 800 — from around compute-optimal (Hoffmann et al., 2022) to heavily over-trained. We generally follow the hyper-parameter setup of Panferov et al. (2025b), although we scale the learning rate for larger models inversely proportional to the model width. We reuse all hyper-parameters (including LR and weight decay) between BF16 baseline and QAT runs. We describe all hyper-parameters in Appendix B.

**Backward pass quantization.** We first validate the accuracy of MS-EDEN for backward pass quantization in isolation. We selectively enable quantization of various tensors of the two backward pass GEMMs, denoted as $E \cdot W$ and $E^T \cdot X$, and measure the final validation loss increase relative to the BF16 baseline. We test the following schemes:

(a) $E \cdot W^T$, $Q(E^T) \cdot Q(X^T)^T$: Quantization of the weight gradient GEMM.

(b) $Q(E) \cdot W$, $E^T \cdot X$: Quantization of the input gradient GEMM *without* weight re-quantization.

(c) $Q(E) \cdot Q(W^T)^T$, $E^T \cdot X$: Quantization of the input gradient GEMM *with* weight re-quantization.

(d) $Q(E) \cdot W$, $Q(E^T) \cdot Q(X^T)^T$: Quantization of both GEMMs *without* weight re-quantization.

(e) $Q(E) \cdot Q(W^T)^T$, $Q(E^T) \cdot Q(X^T)^T$: Quantization of both GEMMs *with* weight re-quantization.

For outlier smoothing, whenever both tensors in a GEMM are quantized, we perform RHT on the inner dimension of

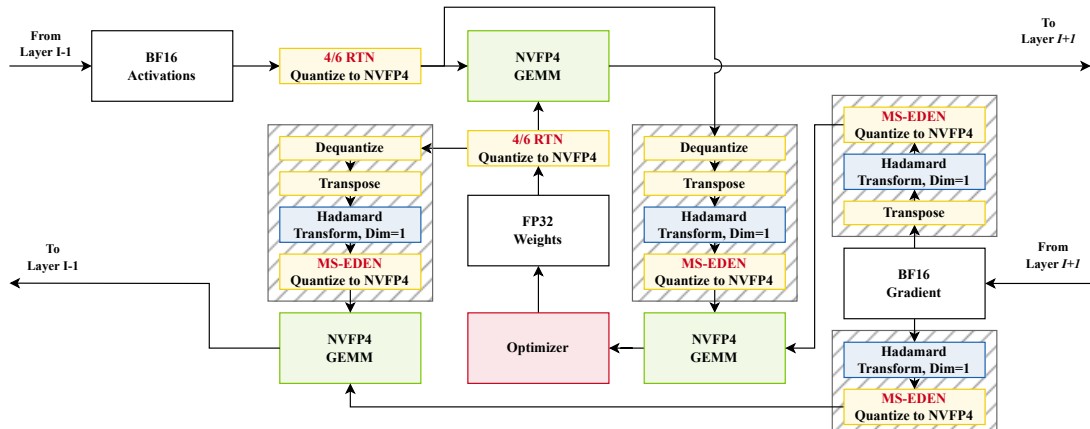

*Figure 3.* Quartet II fully-NVFP4 linear layer computation scheme.

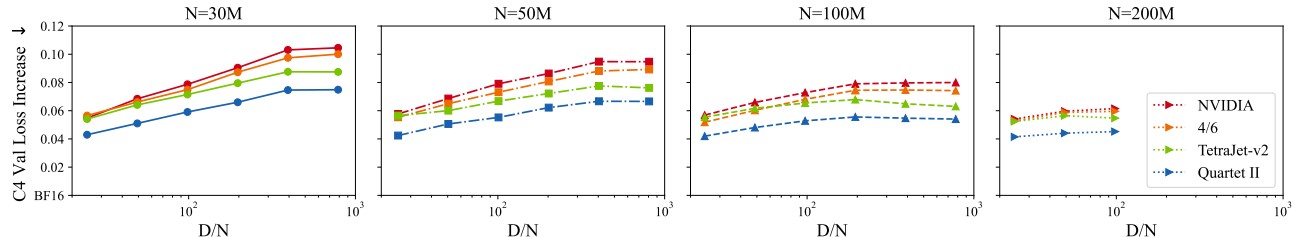

*Figure 4.* Fully-NVFP4 (forward pass and backward pass) C4 Validation Loss Gaps relative to BF16 pre-training for $N$-parameter Llama-2-like LLMs with $D/N$ tokens-per-parameter for Quartet II and baselines.

the GEMM in groups of 128. Naturally, MS-EDEN is incompatible with schemes (b) and (d) as it *requires* weight re-quantization. Nevertheless, we observe that MS-EDEN consistently outperforms SR for each scheme where both are applicable and, more notably, fully-quantized MS-EDEN with weight re-quantization (Figure 1 (e)) outperforms fully quantized SR without weight re-quantization (Figure 1 (d)).

**Forward pass quantization.** Secondly, we validate the effect of square-group-scaling and "4/6" on forward pass quantization in isolation. The results, shown in Figure 2, demonstrate that "4/6" consistently improves both square-group scaling and native-group-scaling weights NVFP4 quantization, as seen by decreasing performance gap vs. BF16. Native weight scales, however, show approximately double the improvement, which aligns with the fact that "4/6" improves both weights and activations quantization there, as opposed to effectively only activations for square-group scaling. This aligns with the quadratic errors in Table 1. Overall, this demonstrates that "4/6" synergizes with native group scales — a novel result we incorporate into Quartet II.

**Full quantization.** Finally, we combine forward pass quantization with backward pass quantization and compare Quartet II against NVIDIA et al. (2025), FourOver-Six (Cook et al., 2025) and TetraJet-v2 (as described in

Section 2) (Chen et al., 2025b). Figure 4 indicates that Quartet II improves consistently w.r.t. both isolated ablations and prior schemes, by at least 20% in terms of loss.

### 6.2. Nanochat Pre-Training

To validate Quartet II at larger scale and on higher-quality data, we provide results for the Nanochat (Karpathy, 2025) training pipeline. It differs from the ablations setup of Section 6.1 in a number of ways: 1) it utilizes the Muon optimizer (Jordan et al., 2024) with WSD LR schedule (Hu et al., 2024), 2) QK-normalization (Henry et al., 2020; Dehghani et al., 2023) and 3) ReLU$^2$ MLP activations (So et al., 2022). Data-wise, Nanochat models are pre-trained on 20 tokens-per-parameter from FineWeb-Edu (Lozhkov et al., 2024) and later fine-tuned on training splits of ARC (Clark et al., 2018), GSM8K (Cobbe et al., 2021), Smol-SmolTalk (Allal et al., 2025) and other smaller datasets. We specify all details in Appendix C.

Similar to Section 6.1, we replace all linear layers with a selected QAT scheme, preserving all training hyper-parameters. We find that Quartet II is stable, and decreases the pre-training loss gap with BF16 by 15-25% relative to existing NVFP4 methods in the pre-training phase, as indicated by validation bits-per-byte's increase over BF16 shown in Figure 5. The zero-shot benchmarks, reported

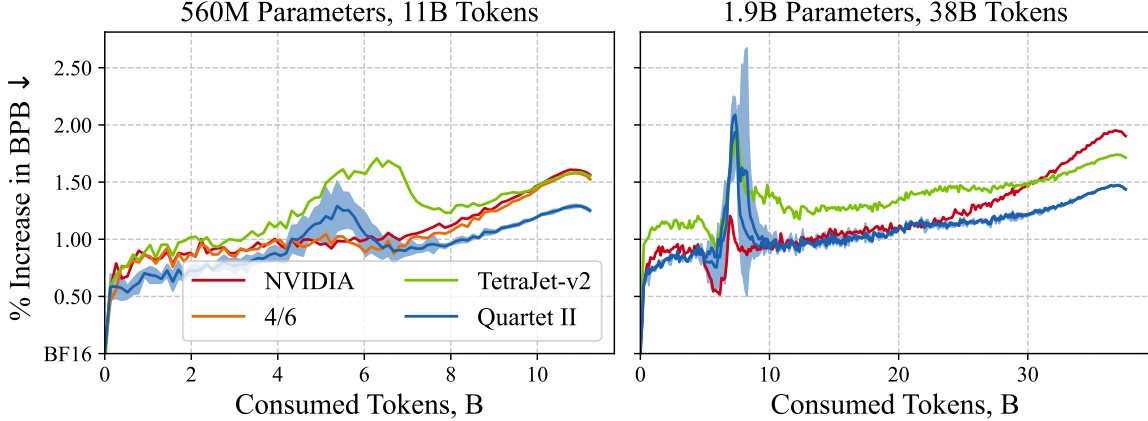

*Figure 5.* Validation loss curves for Nanochat pre-training. Plot show relative increase in bits-per-byte (BPB) w.r.t. BF16 pre-training. Inconsistent loss landscape is observed for both BF16 and QAT around 6B tokens but training always stabilizes later, as seen from reported STD (estimated over 3 restarts) for Quartet II.

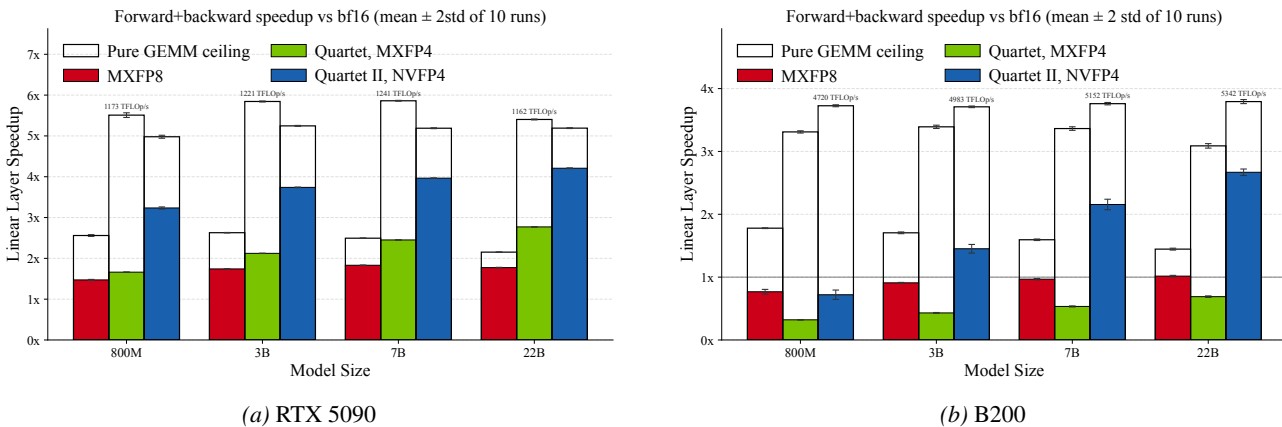

*(a) RTX 5090*  *(b) B200*

*Figure 6.* Linear layer computation speedup over BF16 for training layers characteristic of particular model sizes. Hollow boxes indicate pure matmul speed, and correspondingly visualize the quantization overhead.

after additional mid-training and SFT (Appendix C), show insignificant differences between the QAT methods, probably due to short instruction tuning and small test datasets.

## 7. Kernel Support

**Fused Re-Quantization Kernel.** Hashed regions in Figure 3 indicate roughly which operation can be merged together for efficient execution on GPUs. In practice, however, these operations cannot be performed in a single kernel pass because the global maximum reduction, required for NVFP4 quantization, acts as a global barrier. It has to be performed in a separate kernel, as shown in Figure 7 for the re-quantizing MS-EDEN operation as an example. This doubles the memory bandwidth and matrix multiplication costs, as the entire tensor has to be loaded and rotated twice.

**Post Hoc Range Alignment.** To avoid double loads and rotations, we propose the following format-specific and

hardware-aware implementation heuristic for MS-EDEN: post hoc range alignment for NVFP4 quantization.

In the first kernel, instead of aligning the scales range with a pre-computed AbsMax, we skip the alignment and round scales to E8M3 — an extended range proxy for FP8 represented in BF16. We then divide tensor values by the scales, round to FP4. We refer to this combination of E8M3 scales and FP4 values as *extended-range NVFP4* (ER-NVFP4). We reduce the global absolute maximum after rotation and calculate the EDEN correction factors in the same kernel, removing the need to load and rotate the original tensor twice.

In the second kernel, we load the E8M3 pseudo-scales, as well as the reduced FP32 global maximum, shift the pseudo-scales into the FP8-representable region, apply the EDEN correction and quantize them to FP8 with stochastic rounding, yielding unbiased gradient estimation. The resulting scheme for the re-quantizing MS-EDEN operation, as an example, is shown in Figure 8. Since the second kernel

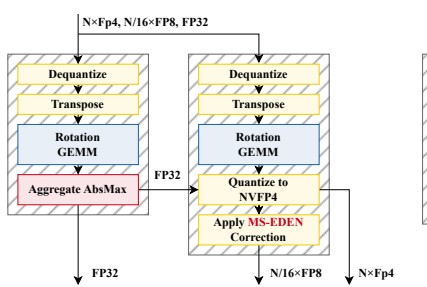
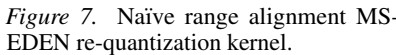

*Figure 7.* Naïve range alignment MS-EDEN re-quantization kernel.

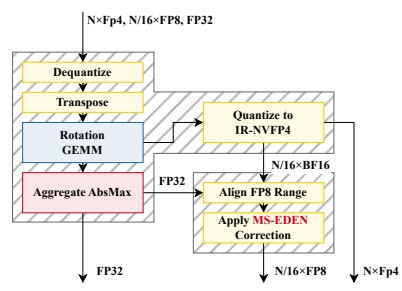

*Figure 8.* Improved post hoc range alignment MS-EDEN re-quantization kernel.

*Table 2.* Global memory (GMEM) bandwidth and GEMM instruction complexities for naïve and post hoc range alignment MS-EDEN re-quantization kernels.

| Kernel: | Naïve | Post hoc |
|---|---|---|
| **Bits moved per element** | | |
| GMEM→SM: | 4.5+4.5 | 4.5+1 |
| SM→GMEM: | 0+4.5 | 5+0.5 |
| **GEMM calls per NVFP4 group** | | |
| `mma.m16n8k16:` | 2 | 1 |

only operates on the scales, it requires substantially less memory movement than the initial quantization, leading to a theoretical bandwidth saving of around 20%, as shown in Table 2, and practical latency of the second kernel being more than 10x less than the first one. We discuss the specific implementation in Appendix D.

**Speedups.** We provide and benchmark custom CUDA kernels tailored for the NVIDIA RTX 5090 GPU and B200 for all three unique Quartet II quantization operations in the backward pass, as well Four Over Six quantization in the forward pass. For the matrix multiplications themselves, we use cuBLAS.

Firstly, to reduce the effect of external factors (e.g., distributed setting, attention implementation, vocabulary size), we report the isolated speedup of linear layer operations. Figure 6 shows these results on a RTX 5090 (1676 TFLOP/s; theoretical speedup $8\times$ and on a B200 (9000 TFLOP/s; theoretical speedup $4\times$). Due to power or thermal constraints, as well as due to matrix shapes, the actual FLOP/s achieved for a pure matmul are below the theoretical speeds, and shown as hollow boxes in the plots. The filled boxes indicate the actual speeds, with the gaps between the two corresponding to the cost of our quantization kernels.

For the RTX 5090, Quartet II achieves more than $4\times$ linear layer speed for large sizes, and still more than $4\times$ even for small sizes, and consistently outperforms the implementation from Quartet (Castro et al., 2025). On the B200, the smaller matrix sizes are entirely dominated by the quantization overhead, and we see actual speedups only starting at 3B, and get up to $2.5\times$ for 22B. Moreover, we demonstrate a more than 1.8x increase over BF16 in real training throughput for 1B LLM pre-training (details in Appendix D).

We validate that MS-EDEN's better guarantees imply better model quality and that the proposed scheme benefits from additional QAT heuristics. The hardware support we provide in the form of CUDA kernels further demonstrates its practical potential.

Even though we see the main contribution of this paper in the application of advanced debiasing techniques to the backward pass of deep learning model training, the experimental verification focused on a relatively narrower problem of pre-training LLMs in the NVFP4 format, which currently is only supported on a single generation of NVIDIA accelerators. From basic statistical properties (i.e., measurably lower MSE on Gaussian source), we expect the approach to generalize to other problems and model architectures too, but doing that might require additional hyper-parameter tuning, such as randomized rotation temporal and spacial granularity.

## 8. Conclusion and Limitations

We leveraged insights from distributed optimization to propose a novel unbiased quantization scheme for microscaling formats called MS-EDEN. Based on it, we propose Quartet II — a computation scheme for NVFP4 LLM pre-training.

## Impact Statement

This paper presents work whose goal is to advance the field of Machine Learning. There are many potential societal consequences of our work, none which we feel must be specifically highlighted here.

## Acknowledgments

We would like to thank Anjulie Agrusa and Tijmen Blankevoort (NVIDIA), for their methodological input and for reviewing the manuscript. Additionally, we would like to thank our contacts at Datacrunch/Verda, Paul Chang and Antonio Dominguez, for hardware support that was essential to this project. Last but certainly not least, we would like to thank Roberto L. Castro for help with efficient NVFP4 matrix multiplication kernels. This work was supported under project ID 40 as part of the Swiss AI Initiative, through a grant from the ETH Domain and computational resources provided by the Swiss National Supercomputing Centre (CSCS) under the Alps infrastructure. This research was funded in whole or in part by the Austrian Science Fund (FWF) 10.55776/COE12.

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

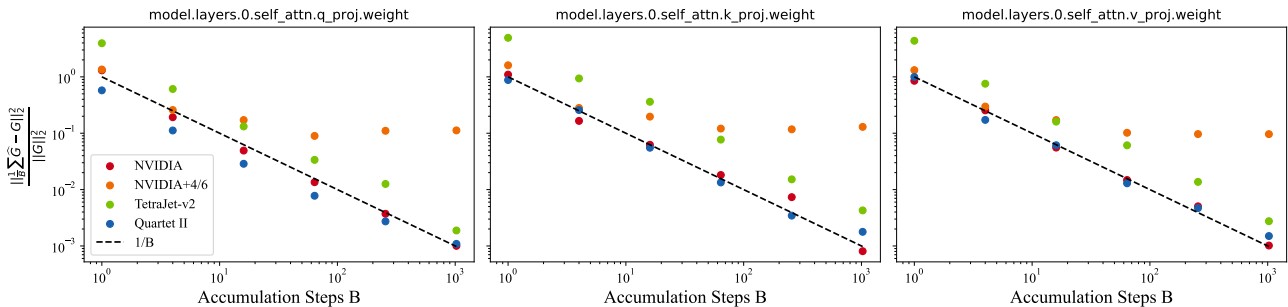

*Figure 9.* Concentration of quantized backward average towards unquantized backward for Quartet II and a number of baselines. Methods parallel to $1/B$ are unbiased. Plateauing methods (NVIDIA+4/6) introduce bias.

## A. Unbiasedness Verification

In this section we verify that Quartet II produces effectively unbiased gradient estimates when applied to backward pass quantization in LLMs.

Firstly, the original EDEN algorithm utilized high-precision unbiasing factors $S$ per every rotation group. Algorithm 1, however, calculates correction factors $S$ per-NVFP4-group (16 elements) and then merges them into FP8 shared exponents via stochastic rounding (SR). Performing unbiasing in smaller groups allows us to reduce the amount of inter-thread communications in GPU kernels when reducing the scalar products for $S$. At the same time, it preserves the unbiasedness argument by considering the larger RHT as a two-level scheme where vectors are first rotated in groups of 16, which yields unbiasedness, and then mixed in groups of 128 to further smooth out the outliers.

Even though EDEN (Vargaftik et al., 2022) only guarantees unbiasedness in the $d \to \infty$ limit and requires rotations to be sampled independently, in practice we make a number of compromises to improve hardware compatibility:

1. We use $d = 128$ to allow efficient rotation on Blackwell GPUs using the `mma.m16n8k16` instruction.

2. We apply identical rotations for every rotation group within a tensor for every micro-batch to reformulate the rotation as simple GEMM. I.e., we re-randomize rotations per-tensor per-micro-batch.

3. We don't perform stochastic rounding on under-flowing FP8 values in MS-EDEN to simplify the bit-manipulation code. This can only affect scales that are at least $\approx 32000$x smaller than the largest scale in each tensor, which makes the effect negligible.

We numerically verify the unbiasedness for LLMs by performing repeated ($B$ times) quantized backward passes over a batch of sample data and calculating the relative quadratic error of the average quantized gradient $\frac{1}{B}\sum \widehat{G}(\omega)$ w.r.t. the reference unquantized gradient $G$. If $\widehat{G}$ is unbiased, i.e., $\mathbb{E}_\omega \widehat{G}(\omega) = G$, the error will decrease to arbitrarily small values as $\sim \frac{1}{B}$ asymptotically, from the Central Limit Theorem. Figure 9 shows that this property holds in practice for the Quartet II implementation with the aforementioned hardware optimization. Additionally, it shows how the gradient estimates produced by NVIDIA (NVIDIA et al., 2025) and TetraJet-v2 (Chen et al., 2025b) are also unbiased, while the application of Four Over Six (Cook et al., 2025) to the backward pass isn't.

For this experiment we used the `Llama-3.2-1B` (Grattafiori et al., 2024) pre-trained model to verify that the unbiasedness is not due to attuning to QAT dynamics. Moreover, in Figure 9 we report error concentration for the attention block 0 - the deepest in the model from the backpropagation perspective.

## B. Llama-Like Hyper-Parameters

We list model-specific hyper-parameters in Table 3 and hyper-parameters shared across all experiments in Table 4.

*Table 3.* Model-specific hyper-parameters used for Llama-like models.

| Hyperparameter | 30M | 50M | 100M | 200M |
|---|---|---|---|---|
| Number of Layers | 6 | 7 | 8 | 10 |
| Embedding Dimension | 640 | 768 | 1024 | 1280 |
| Attention Heads | 5 | 6 | 8 | 10 |
| Learning Rate | 0.0012 | 0.0012 | 0.0009 | 0.00072 |

*Table 4.* Common hyper-parameters used across all model sizes and quantization setups for Llama-like models.

| Hyperparameter | Value |
|---|---|
| Sequence Length | 512 |
| Batch Size | 512 |
| Optimizer | AdamW |
| Learning Rate Schedule | Cosine, 10% warm-up |
| Gradient Clipping | 1.0 |
| Weight Decay ($\gamma$) | 0.1 |
| Number of GPUs | 8 |
| Data Type (optimizer/accumulators) | `FP32` |

## C. Nanochat Details and Extra Evaluation

For our experiments we use the revision of Nanochat indicated by this commit hash:

```
f5425245f99efd4145d2ac71a730af1e96777d6a.
```

At this revision, we focus on two scripts: `speedrun.sh` that trains a 560M parameters model on 11B tokens and `run1000.sh` that trains a 1.9B parameters model on 38B tokens. We first run the `speedrun.sh` script as the unquantized baseline, then add custom QAT support and run it for every tested QAT method. For pre-training, we perform fully-quantized training, i.e., both forward pass and backward pass quantization. For post-training (mid-training and SFT), however, we disable backward pass quantization to get the most out of these very short and data-limited phases. After that, we repeat the process with the `run1000.sh` script.

We report the final normalized pre-training accuracies for Arc-Challenge and Arc-Easy (Clark et al., 2018), PIQA (Bisk et al., 2019), HellaSwag (Zellers et al., 2019) and the CORE-Pretrain score nanochat provides. There, we also report bootstrapped trust intervals (2 standard deviations) w.r.t. re-running the exact same Quartet II pre-training thrice. From them, one can see that the vast majority of the differences between FP4 QAT methods are not statistically significant.

## D. Kernel Benchmarks

### D.1. Linear-Wise Speedups

In Figure 6, we demonstrated FP4 speedups over BF16 for linear layer training. By that, we mean the latency reduction for performing a single forward pass and a single backward pass for a set of layers that would normally be present in a transformer (Vaswani et al., 2023) model of particular size. The actual tensor shapes used for these measurements are presented in Table 6. For these measurements, we use batch size 8 and sequence length 2048. We further assume that the abs-max (for global scale) is provided externally; in practice, the abs-max reduction can be fused into the previous kernel (optimizer for weights, nonlinearity/norm for activations) with negligible performance cost.

In addition to the full speed-ups shown in the main body of the paper, Figure 10 shows the speedup on the forward-pass only. As most of the more expensive quantization kernels are only required during the backward (forward only needs four-over-six rounding), forward-only speedups are much closer to raw matmul speedups.

*Table 5.* Nanochat pre-training final bits-per-byte (BPB) and few-shot metrics for BF16 and a number of FP4 QAT methods.

| Method | BF16 | NVIDIA | 4/6 | TetraJet-v2 | Quartet II | 95% CI |
|---|---|---|---|---|---|---|
| *Nanochat Speedrun: 560M Paramters, 11B Tokens* | | | | | | |
| Val BPB ↓ | 0.7693 | 0.7813 | 0.7809 | 0.7825 | **0.7790** | ±0.0008 |
| Increase over BF16 ↓ | - | 1.56% | 1.51% | 1.72% | **1.26%** | ±0.08% |
| CORE ↑ | 23.5% | 21.3% | 20.9% | 22.5% | 21.9% | ±0.6% |
| ArcC ↑ | 13.0% | 12.5% | 13.3% | 12.7% | 12.4% | ±4.0% |
| ArcE ↑ | 55.6% | 52.4% | 53.6% | 53.1% | 54.2% | ±0.8% |
| PIQA ↑ | 39.4% | 39.1% | 38.7% | 37.8% | 37.8% | ±1.5% |
| HellaSwag ↑ | 31.3% | 29.0% | 29.2% | 28.9% | 29.1% | ±0.6% |
| *Nanochat 1000$: 1.9B Parameters, 38B Tokens* | | | | | | |
| Val BPB ↓ | 0.6925 | 0.7058 | 0.7047 | 0.7044 | **0.7025** | ±0.0002 |
| Increase over BF16 ↓ | - | 1.92% | 1.76% | 1.72% | **1.44%** | ±0.02% |
| CORE ↑ | 34.2% | 32.0% | 31.5% | 32.1% | 32.2% | ±0.2% |
| ArcC ↑ | 26.5% | 25.5% | 26.7% | 25.5% | 25.7% | ±1.6% |
| ArcE ↑ | 64.8% | 62.3% | 63.5% | 62.1% | 63.2% | ±0.6% |
| PIQA ↑ | 50.7% | 50.4% | 49.8% | 48.6% | 38.4% | ±2.6% |
| HellaSwag ↑ | 50.6% | 48.2% | 48.5% | 48.3% | 48.4% | ±0.4% |

*Table 6.* Weight shapes characteristic of Llama-like models of certain sizes as `[in_dim,out_dim]`. We report speedups for these shapes, aggregated as latency for each model size, in Figure 6.

| Layer | 800M | 3B | 7B | 22B |
|---|---|---|---|---|
| QKV | `[2048,6144]` | `[3072,9216]` | `[4096,12288]` | `[6144,18432]` |
| Out | `[2048,2048]` | `[3072,3072]` | `[4096,4096]` | `[6144,6144]` |
| UpGate | `[2048,11264]` | `[3072,16384]` | `[4096,22016]` | `[6144,32768]` |
| Down | `[5632,2048]` | `[8192,3072]` | `[11008,4096]` | `[16384,6144]` |

## D.2. End-to-End Speedups

**RTX 5090.** In addition to the linear layer only, we also run benchmarks with full model training on a single 5090. As this GPU has only 32GB of memory, this limits the maximum model size to using nanochat with a depth of 26, corresponding to 1.1B parameters. When using master-parameters in BF16, this allows for a batch size of 4. At such small sizes, we found it beneficial to fuse the Q, K, and V matrix multiplications into a single kernel call, for both the bf16 baseline and the nvfp4 version.

In this setting, the bf16 baseline achieves a training speed of 28 ktok/s, corresponding to an MFU of 92%. The nvfp4 training reaches a speed of 52 ktok/s, or 185% that of the baseline. Table 7 shows the contribution of different operations to the total runtime. As can be seen, at this model size, about 60% of the time is spent on operations untouched by the FP4 training recipe. This ratio is expected to drastically decrease as model size grows, increasing the usefulness of FP4.

**B200.** In addition to the layer-wise measurements, we also ran some more realistic end-to-end setups. We used an OLMO2 (OLMo et al., 2024) architecture at varying sizes, running for several hundred steps on an 8xB200 server with a global batch size of 524288 tokens per AdamW update step. For model sizes 3.3B, 5.6B, 7.1B, 8.8B, and 11B we get end-to-end speedups over the BF1 baseline of 1.48, 1.58, 1.59, 1.62 and 1.68. While these speedups seem disappointingly small compared to the theoretical 4×, we note that they are generally in line with numbers reported by NVidia [1] for similar setups.

---

[1]https://developer.nvidia.com/blog/using-nvfp4-low-precision-model-training-for-higher-throughput-without-losing-accuracy/

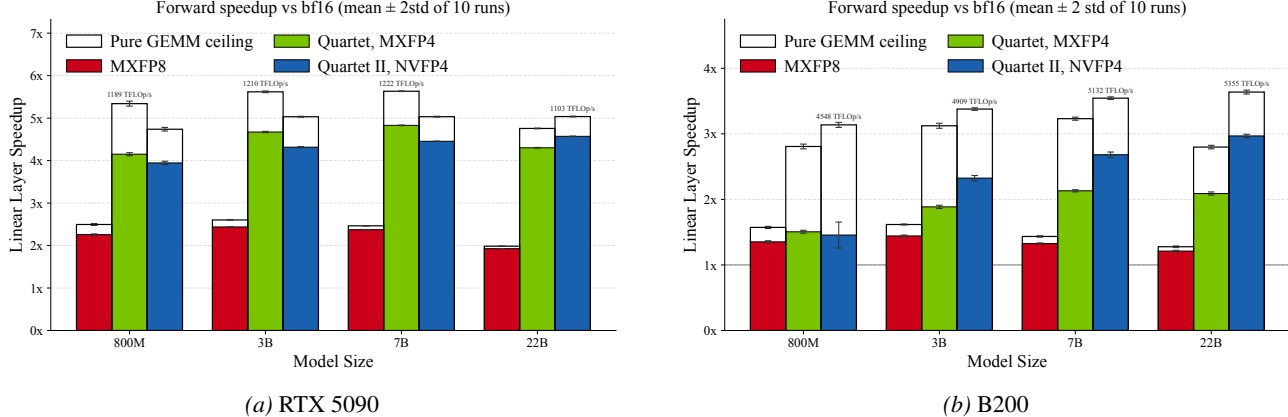

*(a)* RTX 5090                                                          *(b)* B200

*Figure 10.* Linear layer computation speedup over BF16 for training layers characteristic of particular model sizes. Hollow boxes indicate pure matmul speed, and correspondingly visualize the quantization overhead.

*Table 7.* Breakdown of time spent in different kernels (or kernel types, e.g., backward attention is the sum of three different kernels) in the 1.1B parameter model at 8192 tokens per pass.

| | Forward | | | Backward | |
|---|---|---|---|---|---|
| OP | Time [μs] | Fraction | Op | Time [μs] | Fraction |
| FP4 GEMM | 11708 | 24% | FP4 GEMM | 22013 | 21% |
| Attention | 9242 | 19% | Attention | 21500 | 20% |
| RMSNorm | 8498 | 17% | LM-Head | 15734 | 15% |
| LM-Head | 7528 | 16% | RMS-bwd | 12824 | 12% |
| Quantization | 3852 | 8% | Grad Quant. | 10212 | 10% |
| Relu² | 3545 | 7% | Relu²-bwd | 5010 | 5% |
| Abs-Max | 1422 | 3% | Requant | 3532 | 3% |
| Loss | 658 | 1% | Scale Fixup | 1036 | 1% |
| Other | 1672 | 3% | Other | 13639 | 13% |

