# OpenReview forum: "Quartet II: Accurate LLM Pre-Training in NVFP4 by Improved Unbiased Gradient Estimation"
_ICML.cc/2026/Conference — ICML 2026 regular_

### Official Review · Reviewer_oqRh · 2026-03-03

**Soundness:** 3
**Presentation:** 4
**Significance:** 3
**Originality:** 3
**Overall Recommendation:** 5
**Confidence:** 4

**Summary:**

This paper introduces Clover, a NVFP4 capable scheme for pre-training Large Language Models. The authors observe that existing state-of-the-art FP4 training methods(SR) for unbiased gradient estimation inflates quantization error. To mitigate this, the paper proposes MS-EDEN, a novel unbiased quantization routine that shifts the stochasticity from individual values to the microscale factors. Evaluations show that Clover achieves consistently better pre-training loss compared to prior NVFP4 recipes, and custom CUDA kernels customized for Blackwell GPUs demonstrate over 4x speedup relative to BF16 for linear layers.

**Compliance With Llm Reviewing Policy:**

Affirmed.

**Final Justification:**

This work is already solid. I’ll keep my original acceptance decision.

**Key Questions For Authors:**

1. The BPB improvements during pre-training do not translate into significant downstream gains (Appendix C, Table 5). Why does improved gradient estimation fail to differentiate tasks after SFT?
2. Switching from 16×16 to 1×16 scales increases the number of FP8 scales per tensor by 16×, and Clover additionally requires saving forward-pass tensors for backward re-quantization. What is the net memory overhead compared to the NVIDIA baseline that reuses forward-pass weights directly?
3. Appendix C states that backward-pass quantization is disabled during SFT because the phase is "short and data-limited," meaning MS-EDEN is absent during the stage that directly determines Table 5 scores. If low-precision backward passes are too noisy for short training phases, does the same concern apply to the late stages of pre-training where gradient magnitudes are similarly small?

**Limitations:**

yes

**Strengths And Weaknesses:**

**Strengths**
1. The theoretical adaptation of EDEN to microscaling formats is elegant. By shifting the stochastic rounding from 4-bit elements to 8-bit scales, the method provably maintains expected unbiasedness while cutting the high variance typically injected by stochastic rounding.
2. The paper is exceptionally well-structured and clearly written. The progression from theoretical motivation (unbiasedness vs. MSE trade-off) to the proposed algorithm, and finally to kernel implementation, makes the narrative easy to follow.
3. Figure 3's computation graph is an excellent visual summary of the full Clover forward/backward pipeline, making the otherwise complex multi-stage quantization scheme immediately legible to the reader.
4. Training in NVFP4 without catastrophic accuracy degradation is a highly practical and impactful problem for the community. Decreasing the training cost of LLMs addresses a crucial bottleneck in foundation model scaling.
5. While randomized Hadamard transforms (RHT) and group scaling have been explored, combining them into MS-EDEN specifically to sidestep the precision limitations of NVFP4 scales is a highly novel theoretical and systems co-design.

**Weaknesses**
1. The downstream evaluations (Table 5) show no statistically significant differences across FP4 methods, as the authors acknowledge in Appendix C — the confidence intervals at 1.9B scale are too wide (e.g., ArcC ±2.8%). Larger-scale validation would be more convincing.
2. The MSE advantage (Table 1) is measured on synthetic N(0,1) data. Real LLM gradients have heavy tails and outliers — per-layer MSE on actual training data would be more compelling.
3. Backward-pass quantization is disabled during SFT (Appendix C), reverting to BF16 gradients. Since MS-EDEN is a backward-pass technique, it is absent during the phase that directly determines the downstream scores in Table 5.

---

> ### Author Rebuttal · Authors · 2026-03-31
>
> Thank you for the review.
>
> We first answer your direct questions:
>
> ## 1. Significance of Downstream Gains
>
> The evaluation suite of Nanochat simply lacks the resolution needed to reliably distinguish NVFP4 QAT methods. To further demonstrate statistical significance/insignificance, we simplify the setup, conduct additional evaluations and report aggregated performance.
>
>  - To reduce the noise, instead of reporting separate post-SFT benchmarks, we now report the “CORE” ensemble evaluation performed by Nanochat right after the pre-training phase. This ensemble includes a total of 22 0-shot and few-shot benchmarks (including centered Arc Challenge, Arc Easy, PIQA and HellaSwag) and reports their centered average.
>  - To better estimate the standard deviations, instead of reporting Wald intervals derived from test set sizes, we explicitly bootstrap for STD for each benchmark (and final validation BPB) by fully re-running the Clover Nanochat 1.9B pre-training thrice.
>
>
> The table below shows the measured pre-trained Nanochat performance for as well as the bootstrapped standard deviations.
>
> | Method           | Val BPB | CORE  | ArcC  | ArcE  | PIQA  | HellaSwag |
> |:-----------------|:--------|:------|:------|:------|:------|:----------|
> | BF16             | 0.6925  | 0.342 | 0.265 | 0.648 | 0.507 | 0.506     |
> | Clover           | 0.6995  | 0.322 | 0.257 | 0.632 | 0.484 | 0.484     |
> | TetraJet-v2      | 0.7011  | 0.321 | 0.255 | 0.621 | 0.486 | 0.483     |
> | NVIDIA+4/6       | 0.7016  | 0.315 | 0.267 | 0.635 | 0.498 | 0.485     |
> | NVIDIA           | 0.7024  | 0.320 | 0.255 | 0.623 | 0.504 | 0.482     |
> | Bootstrapped STD | 0.0001  | 0.001 | 0.008 | 0.003 | 0.013 | 0.002     |
>
> These new results show that most of the downstream evaluation differences between NVFP4 QAT methods are statistically insignificant, even when averaging over dozens of benchmarks. Meanwhile, Validation BPB shows significant performance differences.
>
> Even though we cannot prove a significant difference in zero-shot accuracy for the NVFP4 QAT methods with the experiments presented here, [Quartet](https://arxiv.org/abs/2505.14669) has shown that lower validation BPB translates to improved downstream performance. From this we conclude that although downstream performance differences are not provably significant, significantly lower validation BPB of Clover implies better downstream performance too.
>
> ## 2. Memory Impact of 16x16 Scaling
>
> With 16×16 scales, the total bitwidth is 4+8/256=4.03bpw – a mere 0.5bpb decrease over 4.5bpw of native NVFP4. Moreover, this saving is further diluted by master weights that are kept in 16- or 32-bit precision, making this 0.5bpw saving unnoticeable. As such, Clover has close to no memory overhead compared to the NVIDIA recipe.
>
> ## 3. Backward Pass Quantization and Gradient Magnitude
>
> From Figure 5, one can see that the performance gap between BF16 and QAT actually decreases during the final few percent of the training duration, indicating that small gradient magnitude is not a problem for NVFP4 QAT. The decision to disable backward-pass quantization for Nanochat SFT was a heuristic to reduce the downstream evaluation variance. The new “CORE Metric” evaluations above, however, show that downstream evaluation variance is too large to reliably distinguish the methods even before the SFT phase, rendering the choice inconsequential.
>
> ## Regarding "Evaluating the MSE on real tensors"
>
>
> We reported MSE on N(0,1) data because almost all backward-pass quantization methods employ randomized rotations. These rotations inherently transform any gradient distribution into a centered Gaussian, making our synthetic results universally applicable.
>
> The only exception is the NVIDIA scheme, which uses Stochastic Rounding (SR) without rotations for the gradient tensor. To directly address your concern, we measured the relative per-tensor MSE (relMSE) on real training gradients from a 1.9B Nanochat model:
>
> | Method      | relMSE (real gradients, 1e-3) | relMSE (N(0,1), 1e-3) |
> |:------------|:------------------------------|:-|
> | SR          | 22.8±1.8                      | 23.5 |
> | MS-EDEN     | 9.4±0.4                       | 9.8 |
>
> > Note: Standard deviations are calculated across model layers
>
> These real-tensors results closely mirror our synthetic findings re-reported from Table 1, confirming MS-EDEN's significant advantage.

---

> > ### Author Rebuttal · Reviewer_oqRh · 2026-04-02
> >
> > Thanks for the respons. Everything is clear now.

---

### Official Review · Reviewer_Sm9D · 2026-03-12

**Soundness:** 2
**Presentation:** 2
**Significance:** 3
**Originality:** 2
**Overall Recommendation:** 4
**Confidence:** 2

**Summary:**

This paper presents Clover, a fully NVFP4 training scheme for large language models designed for NVIDIA Blackwell GPUs. It targets low precision pre training, with the goal of reducing compute cost while preserving convergence. The main contribution is MS-EDEN, an unbiased quantization method for microscaling FP4 that adapts EDEN style bias correction by stochastically rounding correction factors into FP8 group scales instead of individual FP4 values. This yields more than 2 times lower MSE than standard stochastic rounding while maintaining unbiasedness. Clover uses MS-EDEN in the backward pass and combines RTN quantization with the Four Over Six scale heuristic in the forward pass. Experiments show at least 20 percent smaller C4 validation loss gaps than prior NVFP4 recipes, from small scale ablations up to end to end Nanochat pre training with 1.9B parameter models. The paper also includes hardware aware CUDA kernels that deliver up to 4.2 times speedup over BF16 for linear layers.

**Compliance With Llm Reviewing Policy:**

Affirmed.

**Final Justification:**

All of my questions have been solved. I have increased my score to 4.

**Key Questions For Authors:**

1. The main evidence for Clover’s advantage over stochastic rounding comes from 30M to 200M parameter ablations, with the largest end to end experiment at 1.9B parameters and 38B training tokens. Do the authors have empirical or theoretical evidence that the MSE reduction from MS-EDEN continues to produce a meaningful validation loss advantage at larger scales, such as 7B parameters and beyond? In particular, as model size and batch size grow, optimization may become less sensitive to per step gradient noise, which could reduce the practical impact of improved gradient estimation.

2. Table 5 shows that Clover achieves the best pre training BPB in both Nanochat settings, yet this advantage does not consistently carry over to downstream evaluation after SFT. At 1.9B scale, Clover is worse than TetraJet-v2 on ARC-Challenge, ARC-Easy, and MMLU despite its lower pre training BPB. The paper attributes this to noise from short instruction tuning and small benchmark sizes, but does not report confidence intervals for the 1.9B results. Could the authors either provide uncertainty estimates for these downstream scores or offer a more concrete explanation for why a 0.28 percent absolute BPB gain fails to produce consistent downstream improvement?

**Limitations:**

No. The Impact Statement is perfunctory and does not adequately discuss.

**Strengths And Weaknesses:**

**Strengths**

**1. Clear theory to support empirical findings.**
The paper provides a principled and technically nontrivial adaptation of EDEN to the NVFP4 microscaling setting. It clearly explains why the original EDEN correction is incompatible with coarse FP8 group scales and introduces stochastic rounding of the group scales as a concrete fix.

**2.  Practical hardware-aware implementation.**
Unlike several concurrent works that report only emulated or software-simulated results, Clover is accompanied by custom CUDA kernels for the NVIDIA RTX 5090 (Blackwell). The post-hoc range alignment optimization (Section 7, Figures 7–8) is a substantive engineering contribution that reduces both memory bandwidth (approximately 20% theoretical saving, Table 2) and GEMM calls per NVFP4 group from 2 to 1.

**Weaknesses**

**1. Limited scale of the training experiments.**
The paper positions Clover as a solution for large scale LLM training, but the experimental evidence remains relatively modest for that claim. The largest end to end result is a 1.9B parameter model trained on 38B tokens, while the main ablations in Figures 1, 2, and 4 are limited to models up to 200M parameters. This leaves a substantial gap between the scope of the claims and the scale of the evaluation. Since quantization behavior can change non-monotonically with model size, especially in the transition to truly production scale regimes such as 7B to 70B parameters, it is unclear whether the reported advantages would remain as strong at larger scales.

**2. Limited evidence that pre training gains translate to downstream quality.**
Although Clover shows clear improvements in pre training metrics, the downstream evaluation does not demonstrate a correspondingly consistent advantage. In Table 5, the FP4 QAT methods perform similarly after supervised fine tuning, and in the 1.9B setting Clover is actually worse than TetraJet-v2 on several benchmarks, including ARC-Challenge, ARC-Easy, and MMLU. The

---

> ### Author Rebuttal · Authors · 2026-03-31
>
> Thank you for the review.
>
> First, we present pre-training results for even larger models to address your hesitation about whether the performance improvement persists. We pre-trained 3B and 7B Llama-like models on 32B and 64B tokens respectively. For 3B, we pre-trained a Clover variant, a TetraJet-v2 variant, chosen as the most potent baseline on the 1.9B model, and a BF16 variant. In terms of final loss, Clover showed an 0.83% increase over BF16, while TetraJet-v2 showed a 1,02% increase. For 7B, we only ran our own method with real NVFP4 kernels. That run also showed stable training dynamics, as demonstrated by training loss curves attached: https://freeimage.host/i/B2sAVXn
>
> Running the baselines for the 7B model, as well as scaling further, would be prohibitively expensive, as even the efficient NVFP4 pre-training with the kernels we implemented already took around 4 compute days on an 8xB200 node (around $4K by today’s prices), and some of the baselines (TetraJet-v2, 4/6) don not even provide training kernels.
>
> We hope that this proves that our method’s advantage scales to “production scale regimes” that, as you defined them, begin at 7B models.
>
>
> Next, we answer your direct questions:
>
> ## 1. Larger Models
>
> From a mathematical perspective, MSE of the gradient estimation is the quantity that determines convergence rate bound for gradient-based methods ([Bottou et al.](https://arxiv.org/abs/1606.04838), [Alistarh et al.](https://proceedings.neurips.cc/paper/2017/hash/6c340f25839e6acdc73414517203f5f0-Abstract.html)). From a practical standpoint, [Tseng et al.](https://arxiv.org/abs/2502.20586) tested specifically FP4 backward pass quantization for models up to 7B parameters. The observed performance gaps between FP4 and BF16 persisted on these scales, indicating that the gradient errors specifically continue to matter at that scale, and one would expect that reducing them would naturally improve performance. Moreover, we now present a novel ablation: Backward-pass-only QAT for 1.2B Nanochat models.
>
> | Backward Method | Validation BPB |
> |:----------------|:---------------|
> | BF16            | 0.7183         |
> | Clover          | 0.7207         |
> | TetraJet-v2     | 0.7236         |
> | NVIDIA          | 0.7225         |
>
> These results demonstrate the higher fidelity of MS-EDEN, consistent with Figure 1 (d,e), while scaling 6x in terms of model size.
>
> ## 2. Downstream Performance
>
> The evaluation suite of Nanochat simply lacks the resolution needed to reliably distinguish NVFP4 QAT methods. To further demonstrate statistical significance/insignificance, we simplify the setup, conduct additional evaluations and report aggregated performance.
>
>  - To reduce the noise, instead of reporting separate post-SFT benchmarks, we now report the “CORE” ensemble evaluation performed by Nanochat right after the pre-training phase. This ensemble includes a total of 22 0-shot and few-shot benchmarks (including centered Arc Challenge, Arc Easy, PIQA and HellaSwag) and reports their centered average.
>
> - To better estimate the standard deviations, instead of reporting Wald intervals derived from test set sizes, we explicitly bootstrap for STD for each benchmark (and final validation BPB) by fully re-running the Clover Nanochat 1.9B pre-training thrice.
>
> The table below shows the measured pre-trained Nanochat performance for as well as the bootstrapped standard deviations.
>
> | Method           | Val BPB | CORE  | ArcC  | ArcE  | PIQA  | HellaSwag |
> |:-----------------|:--------|:------|:------|:------|:------|:----------|
> | BF16             | 0.6925  | 0.342 | 0.265 | 0.648 | 0.507 | 0.506     |
> | Clover           | 0.6995  | 0.322 | 0.257 | 0.632 | 0.484 | 0.484     |
> | TetraJet-v2      | 0.7011  | 0.321 | 0.255 | 0.621 | 0.486 | 0.483     |
> | NVIDIA+4/6       | 0.7016  | 0.315 | 0.267 | 0.635 | 0.498 | 0.485     |
> | NVIDIA           | 0.7024  | 0.320 | 0.255 | 0.623 | 0.504 | 0.482     |
> | Bootstrapped STD | 0.0001  | 0.001 | 0.008 | 0.003 | 0.013 | 0.002     |
>
> These new results show that most of the downstream evaluation differences between NVFP4 QAT methods are statistically insignificant, even when averaging over dozens of benchmarks. Meanwhile, Validation BPB shows significant performance differences.
>
> The broader difference between NVFP4 QAT and BF16 is significant either way, implying that the broader dependence between validation BPB and downstream performance still holds, as also shown for QAT by [Castro et al.](https://arxiv.org/abs/2505.14669).
>
> From this we conclude that although downstream performance differences are not provably significant, significantly lower validation BPB of Clover implies better downstream performance too.
>
> **Regarding the Impact Statement,**  will update it to summarize the limitations discussed throughout the paper.

---

> > ### Author Rebuttal · Reviewer_Sm9D · 2026-04-01
> >
> > Thank you for the rebuttal. All of my questions have been solved. I have increased my score to 4.

---

### Official Review · Reviewer_t5hF · 2026-03-12

**Soundness:** 4
**Presentation:** 3
**Significance:** 3
**Originality:** 3
**Overall Recommendation:** 5
**Confidence:** 3

**Summary:**

This paper addresses the problem of accurate LLM pre-training using the NVFP4 microscaling format, which is natively supported on NVIDIA Blackwell GPUs. The authors identify that the prevailing approach for unbiased gradient estimation unnecessarily inflates quantization variance.
Their primary contribution is MS-EDEN, a new unbiased quantization primitive for microscaling formats that reduces MSE by relocating stochasticity from individual FP4 values to the FP8 group micro-scales, leveraging randomized Hadamard transforms  for bias correction. Built on MS-EDEN, the paper proposes Clover, a fully-NVFP4 linear layer training graph that pairs a high-capacity RTN+4/6 forward pass with an unbiased MS-EDEN backward pass. The authors provide an unbiasedness proof, ablations on Llama-family models, end-to-end validation at up to 1.9B parameters on 38B tokens, and custom CUDA kernels achieving up to 4.2x speedup over BF16.

**Compliance With Llm Reviewing Policy:**

Affirmed.

**Key Questions For Authors:**

(1)	In MS-EDEN, the authors mention the clipping factor s. Did the author use a static s across all layers and training stages for the 1.9B Nanochat model? Did you observe any instability spikes that required dynamic tuning of s?

(2)	The unbiasedness verification in Appendix A is conducted on a pre-trained Llama-3.2-1B model rather than during actual QAT training. Does the unbiasedness property hold consistently throughout training, particularly in the early stages when gradient magnitudes are large and distributions shift rapidly?

(3)	Clover focuses heavily on the GEMMs. In the 1.9B training, were the optimizer states (e.g., Adam momentum) kept in FP32/BF16? If optimizer states were quantized to lower precision, would the unbiasedness guarantee of MS-EDEN still hold, or would additional corrections be required?

**Limitations:**

The authors have implicitly addressed some limitations in the appendix, such as the moderated end-to-end training speedup and the lack of statistically significant improvements in downstream zero-shot tasks. However, the authors are strongly encouraged to move a summarized version of these limitations into a dedicated section in the main text. Furthermore, the authors should explicitly acknowledge the hardware lock-in (reliance on specific NVFP4 micro-scaling blocks) and the need for future validation on ≥7B scale models to make the paper perfectly well-rounded.

**Strengths And Weaknesses:**

Strengths

(1)	The core idea of MS-EDEN is well-motivated. By recognizing that element-wise stochastic rounding on 4-bit values inflates variance unnecessarily, the authors shift the stochasticity to the higher-precision FP8 micro-scales, while keeping the FP4 values deterministic via Round-to-Nearest.

(2)	The paper provides formal mathematical proofs and empirical verification to demonstrate that the proposed approach strictly preserves the unbiasedness required for stable gradient estimation.

(3)	The proposed "Post Hoc Range Alignment" kernel avoids double memory loads during the RHT, and custom CUDA kernels on RTX 5090 demonstrate up to 4.2x speedup over BF16, making the method practically deployable.

(4)	The empirical evaluation is comprehensive. The paper compares Clover against the relevant recent baselines, including NVIDIA's official NVFP4 recipe, TetraJet-v2, and FourOverSix.


Weaknesses

(1)	The paper reports a 2.4x end-to-end training speedup at 1.1B parameters, significantly below the 4.2x linear layer speedup, with Table 7 showing ~60% of runtime spent on operations unaffected by FP4. While the authors note this gap is expected to narrow at larger scales, no quantitative projection or larger-scale throughput measurement is provided to support this claim.

(2)	The method successfully lowers pre-training validation loss gap. However, as noted in Appendix C, the zero-shot downstream task performance (Table 5) shows no statistically significant differences among the evaluated FP4 methods.

---

> ### Author Rebuttal · Authors · 2026-03-31
>
> Thank you for your input.
>
> We first answer your direct questions:
>
> ## (1): Clipping Factor $s$
>
> The static factor $s=0.93$ found as minimizing MSE over MS-EDEN quantization of N(0,1) samples was used in all invocations of MS-EDEN in all of our experiments. It required no further tuning. We believe that Nanochat spikes bear no relation to it as they are observed for BF16 pre-training as well, which lacks all and any QAT-related hyper-parameters.
>
> ## (2): Unbiasedness Verification
>
> We specifically chose Llama-3.2-1B for unbiasedness verification as a more informative measurement than our own models trained with QAT. The logic being that models trained with noisy gradient estimations might learn to adapt their structure to this noise, leading to unbiasedness findings not extrapolating to other models or out-of-domain data samples. By picking an unrelated pre-trained model we believe we provide a more convincing unbiasedness demonstration. From a mathematical perspective, the unbiasedness is guaranteed regardless of training stage. In practice, early training shows the lowest performance gaps between BF16 training and QAT. This can be seen, for example, in Figure 5.
>
>
> ## (3): Optimizer State Quantization
>
> The unbiasedness guarantee of MS-EDEN still holds in a sense that the accumulated gradients are unbiased estimates of high-precision-computations gradients. Even though we demonstrate stable performance with AdamW and Muon optimizers, the broader question of how these gradients are used after their accumulation (i.e., optimizer steps and momentum updates) is not the focus of this paper. Nanochat, for example, uses FP32 master weights and optimizer states. Generally, it is known that at least the master weights can be kept in BF16 as long as stochastic rounding is utilized for weight updates ([Ozkara et al.](https://arxiv.org/abs/2502.20566)), circling back to the usefulness of unbiasedness.
>
> Addressing your other concerns:
>
> ## “End-to-end training speedup”
>
> Below we provide end-to-end speedup measurements for larger models, put them into context of expected speedups and provide reasoning for why we expect them to scale favorably with model size.
>
> The scale of models for which we measure the speedup end-to-end was severely limited by the fact that we had to conduct measurements on a 5090 GPU that only has 32Gb of DRAM. Now, having additionally implemented kernels for efficient B200 execution, we are presenting end-to-end speedup measurements for models up to 11B parameters on 8xB200:
>
> | Model Size  | B200 E2E Speedup   |
> | :---------- | :----------------- |
> | 3.3B        | x1.49              |
> | 5.6B        | x1.58              |
> | 7.1B        | x1.59              |
> | 8.8B        | x1.62              |
> | 11B         | x1.68              |
>
> These speedups align perfectly with those [advertised by NVIDIA themselves](https://developer.nvidia.com/blog/using-nvfp4-low-precision-model-training-for-higher-throughput-without-losing-accuracy/), where they promise an x1.59 speedup when switching from BF16 to NVFP4 training for an 8B dense transformer model. One would expect the speedups to increase even further for larger models. To achieve this, however, the use of complex model sharding strategies is needed, which puts design of such systems out of the scope of this paper.
>
> As for the cost of non-NVFP4 operations, it decreases asymptotically with increasing hidden dimension $d$ (e.g., attention, normalizations and logits calculation scale as $O(d)$ while NVFP4 GEMMs scale as $O(d^2)$ under fixed sequence length pre-training). Additional tricks can be employed to further reduce their cost, such as the use of SSMs and sparse attention for some layers, as utilized by NVIDIA for their [Nemotron](https://research.nvidia.com/labs/nemotron/files/NVIDIA-Nemotron-3-Super-Technical-Report.pdf) models pre-trained with NVFP4.
>
> ## “Zero-shot downstream task performance”
>
> As a similar point was raised by Reviewer `oqRH`, we kindly direct you to our detailed response in that thread (under the heading "Significance of Downstream Gains"). There, we explain why we believe that the evaluation suite of Nanochat lacks the resolution needed to reliably distinguish NVFP4 QAT methods and why "Validation BPB" is a robust and reliable metric to base our finding on.
>
> ## Impact Statement Update
>
> We will update it to summarize the limitations discussed throughout the paper and further discuss the hardware specification of our method.

---

> > ### Author Rebuttal · Reviewer_t5hF · 2026-04-03
> >
> > The authors have generally addressed the raised questions.

---

### Official Review · Reviewer_GVpG · 2026-03-13

**Soundness:** 2
**Presentation:** 3
**Significance:** 3
**Originality:** 2
**Overall Recommendation:** 4
**Confidence:** 4

**Summary:**

The paper proposes Clover, a NVFP4 pre-training method of large language models. The key idea is a new unbiased quantizer MS-EDEN, which improves gradient estimation compared to standard stochastic rounding by moving randomness from individual values to microscaling factors. Clover combines this improved backward-pass quantization with a forward-pass quantization strategy using four-over-six. Experiments on Llama-style models and Nanochat training runs show that Clover reduces the accuracy gap relative to BF16 training while enabling up to 4× speedup in linear layers.

**Compliance With Llm Reviewing Policy:**

Affirmed.

**Final Justification:**

Solid paper. Recommend for acceptance.

**Key Questions For Authors:**

N/A

**Strengths And Weaknesses:**

Strength:
1. MS-EDEN provides lower quantization error than stochastic rounding while preserving unbiased gradients.
2. The paper includes GPU kernels and hardware-aware optimizations, showing practical speedups on Blackwell GPUs.
3. Applying stochastic rounding to the quantized matrix introduces a large overhead, without hardware-level support. This paper avoids this by migrating the overhead to a smaller scaling factor.

Weaknesses:
1. The importance of unbiasedness [1] and hadamard transformation [2] has been discussed for a long time in the low-precision training domain, making the paper's contribution smaller.
2. Table 1 should be measured using real activation and weight tensors during pretraining, not fake tensors. Meanwhile, the authors should report the MSE for multiple precision choices (INT8 [3], FP8) and multiple quantization granularities (1 * 16, 1 * 128, 1 * N [4]) to provide a better understanding of this problem.
3. Experiments mainly involve models up to ~1.9B parameters, leaving uncertainty about behavior for larger-scale models (at least 7B).
4. Applying NVFP4 training to B200/B300 might be challenging due to the lack of CUDA cores, limiting the potential speedup gain.
5. In Figure 6, the authors should also explicitly present the weight matrix size instead of only reporting the model size

[1] Oscillation-Reduced {MXFP}4 Training for Vision Transformers

[2] Quartet: Native fp4 training can be optimal for large language models

[3] Jetfire: Efficient and Accurate Transformer Pretraining with INT8 Data Flow and Per-Block Quantization

[4] Stable and low-precision training for large-scale vision-language models

---

> ### Author Rebuttal · Authors · 2026-03-31
>
> Thank you for your input. We now address your concerns in their original order:
>
> ## 1. “The importance of unbiasedness”:
>
> The importance of unbiasedness is indeed common knowledge in QAT literature, and Hadamard rotations have have been utilized before. However, we are the first to **(a)** achieve LLM gradient estimation unbiasedness without utilizing per-element stochastic rounding (which induces higher variance) and **(b)** use randomized rotations for unbiasedness as opposed to using them for outlier smoothing. This presents a novel fusion of the two concepts, and we were surprised to find that it is possible to obtain close to optimal (RTN) MSE while maintaining unbiasedness on the backward pass.
>
> ## 2. “Table 1 .. real activation and weight tensors”:
>
> We reported MSE on normal data in Table 1 because it is meant to compare backward pass quantization methods, almost all of which employ randomized rotations. These rotations inherently transform any gradient distribution into a centered Gaussian, making our synthetic results universally applicable.
>
> The only exception is the NVIDIA scheme, which uses Stochastic Rounding (SR) without rotations for gradient tensors. To address your concern, we measured the relative per-tensor MSE (relMSE) on real training gradient tensors from a 1.9B Nanochat model:
>
> | Method      | relMSE (real gradients, 1e-3) | relMSE (N(0,1), 1e-3) |
> |:------------|:------------------------------|:-|
> | SR          | 22.8±1.8                      | 23.5 |
> | MS-EDEN     | 9.4±0.4                       | 9.8 |
>
> > Note: Standard deviations are calculated across model layers
>
> These real-tensors results closely mirror our synthetic findings re-reported from Table 1, confirming MS-EDEN's significant advantage.
>
> As for forward pass specifics and alternative data formats, they were not the focus of this work. For finer description of quantization error of weights and activations under microscaling formats, the effect of its parameters such as granularity and its effect on LLMs, please refer to the comprehensive studies by [Egiazarian et al.](https://arxiv.org/abs/2509.23202) and [Chen et al.](https://arxiv.org/abs/2510.25602).
>
> ## 3. “Larger-scale models”:
>
> To address your concerns, we pre-trained 3B and 7B Llama-like models on 32B and 64B tokens respectively. For 3B, we pre-trained a Clover variant, a TetraJet-v2 variant, chosen as the most potent baseline on the 1.9B model, and a BF16 variant. In terms of final loss, Clover showed an 0.83% increase over BF16, while TetraJet-v2 showed a 1,02% increase. For 7B, we only ran our own method with real NVFP4 kernels. That run also showed stable training dynamics, as demonstrated by training loss curves attached: https://freeimage.host/i/B2sAVXn
>
> Running the baselines for the 7B model, as well as scaling further, would be prohibitively expensive, as even the efficient NVFP4 pre-training with the kernels we implemented already took around 4 compute days on an 8xB200 node (around $4K by today’s prices), and some of the baselines (TetraJet-v2, 4/6) do not even provide training kernels.
>
> We hope this clarifies the uncertainty about behavior for larger-scale models.
>
> ## 4. “B200/B300”:
>
> Even though potential speedups on B200 and B300 differ from those on 5090, both GPUs provide extensive hardware support for the NVFP4 format and noticeable performance gains from it can still be achieved. To support this claim, we now include the aggregate speedup measurements for the kernels we implement and tuned for the B200 and B300 Blackwell accelerators (same setup as Figure 6):
>
> | Model Size  | 5090 | B200 | B300 |
> | :---------- | :--- | :--- | :--- |
> | 3B          | x3.9 | x1.4 | x2.2 |
> | 7B          | x4.1 | x2.2 | x2.6 |
> | 22B         | x4.4 | x2.7 | x2.8 |
> | 52B         | x5.0 | x3.0 | x3.0 |
>
> The exact speedups vary widely because of hardware specifications, but we hope that the inclusion of kernels for datacenter-grade GPUs would motivate wider adoption of our method.
>
> ## 5.  “In Figure 6 … weight matrix size”:
>
> The exact shapes are reported in Table 6. Figure 6 presents their aggregate speedups for brevity and readability. We will expand Table 6 to include per-shape measurements.

---

> > ### Author Rebuttal · Reviewer_GVpG · 2026-04-03
> >
> > I maintain my score and recommend acceptance.

---

### Decision · Program_Chairs · 2026-04-30

**Decision:**

Accept (regular)

**Comment:**

### **Summary of Contributions**
The paper proposes MS-EDEN, an unbiased quantization primitive designed for the NVFP4 microscaling format. By shifting stochastic rounding from individual 4-bit values to higher-precision FP8 group micro-scales, the method reduces variance and quantization error compared to standard element-wise stochastic rounding. MS-EDEN is integrated with a "Four Over Six" forward-pass heuristic to form Clover, a fully-NVFP4 linear-layer training graph for Large Language Models. The authors provide custom CUDA kernels for NVIDIA Blackwell GPUs, demonstrating up to a 4.2x speedup over BF16 for linear layers.

### **Decision Reasoning**
The submission received a positive consensus from the reviewers .

* **Strengths:** The reviewers agreed that the theoretical adaptation of EDEN to microscaling formats is mathematically sound and elegant. The hardware-aware implementation, specifically the "Post Hoc Range Alignment" which avoids double memory loads, was highlighted as a substantial engineering contribution that makes the method practically deployable.
* **Rebuttal Impact:** The authors effectively resolved the primary weaknesses raised during the review phase:
    * **Evaluation Scale:** Reviewers noted the initial 1.9B parameter limit was modest for claiming "large-scale" utility. In response, the authors provided additional pre-training results for 3B and 7B Llama-like models, demonstrating stable training dynamics at larger scales.
    * **End-to-End Speedup:** Addressing concerns about the gap between isolated linear layer speedups and end-to-end performance, the authors provided benchmarks on 8xB200 nodes showing end-to-end speedups scaling favorably up to 1.68x for 11B models.
    * **Downstream Performance:** Reviewers questioned why improved Validation BPB did not consistently translate to statistically significant zero-shot downstream task improvements. The authors performed a bootstrap analysis to estimate standard deviations, proving that downstream evaluation variance at this scale is too large to reliably distinguish NVFP4 QAT methods, solidifying Validation BPB as the more robust metric.
    * **Real vs. Synthetic Data:** The authors provided relative MSE measurements on real training gradients from a 1.9B model, confirming that MS-EDEN (9.4) maintains a significant advantage over standard Stochastic Rounding (22.8) on actual heavy-tailed distributions.

### **Conclusion**
The paper provides a well-motivated, theoretically justified, and empirically validated approach to NVFP4 pre-training. The custom hardware-aware kernels and the robust rebuttal addressing scalability and metric reliability strengthen the submission significantly. The paper is a clear accept.